# Organophotoredox-Catalyzed Stereoselective Synthesis of Bicyclo[3.2.0]heptanes via [2+2] Photocycloaddition

**DOI:** 10.3390/molecules30102090

**Published:** 2025-05-08

**Authors:** Tommaso Benettin, Simonetta Resta, Alessandra Forni, Laura Raimondi, Alessandra Puglisi, Sergio Rossi

**Affiliations:** 1Dipartimento di Chimica, Università degli Studi di Milano, Via Golgi 19, 20133 Milano, Italylauramaria.raimondi@unimi.it (L.R.);; 2Institute of Chemical Sciences and Technologies (SCITEC) “Giulio Natta”, National Research Council (CNR), 20133 Milano, Italy; alessandra.forni@scitec.cnr.it

**Keywords:** photocatalysis, [2+2] photocycloaddition, Eosin Y, stereoselective synthesis, aryl bis-enones, bicyclo[3.2.0]heptanes, DFT analysis

## Abstract

The stereoselective synthesis of bicyclo[3.2.0]heptanes via an anion radical [2+2] photocycloaddition of aryl bis-enone derivatives was investigated. By employing chiral oxazolidinone auxiliaries bound to aryl bis-enone substrates, enantioenriched, highly substituted bicyclo[3.2.0]heptanes have been synthesized. The reaction, mediated by Eosin Y and promoted by LiBr under visible light irradiation, has been studied both experimentally and computationally to elucidate the mechanism and stereoselective outcomes. The process proceeds via a *syn*-closure pathway, leading to the formation of the corresponding *cis-anti* diastereoisomers as major products isolated and characterized by X-ray analysis; DFT calculations provided useful insights and computational support which allow a plausible reaction mechanism to be proposed that agrees with the collected experimental data.

## 1. Introduction

The introduction of light as a reagent in organic chemistry has enabled novel reactivities and significantly contributed to the development of more sustainable and environmentally friendly methodologies. In this context, the use of visible light (>370 nm) combined with photocatalysts has enhanced reaction efficiency, reduced side reactions, and promoted the synthesis of products that would otherwise be inaccessible through conventional thermal activation [1,2,3,4].

Among all these examples, the development of the photocycloaddition of unsaturated enones for the synthesis of highly functionalized cycloalkanes has attracted increasing attention, since it allowed easy access to highly functionalized (bi)cyclic compounds which have found applications in drug design, pharmaceuticals, and materials science [5,6]. After the seminal works of Krische related to the diastereoselective cobalt catalyzed [2+2] cycloaddition of aryl bis-enones promoted by single electron transfer (SET) [7], or by exposure of bis-enones to a chrysene anion radical [8], a photochemical approach was developed by Yoon [9,10,11] and Zeitler [12] where the cobalt catalyst was replaced with a ruthenium photocatalyst and an organic dye, respectively. More recently, our group also contributed to the field by developing a stereoselective anion radical [2+2] cycloaddition of aryl bis-enones, enabling precise control over the stereoselectivity depending on whether the reaction is performed under batch or flow conditions [5]. In addition, the mechanism usually invoked for this type of transformation was fully validated by DFT calculations [13]. Examples involving the use of supported photocatalysts on silica [14] and Merrifield resin [13] have also been reported (Figure 1).

Although the stereoselective [2+2] photocycloaddition of aryl bis-enones has been extensively studied, allowing high levels of diastereoselectivity to be achieved, no example of an enantioselective version of such transformation was reported. This is primarily due to the challenges associated with the use of chiral catalytic systems in photochemical transformations, as well as the short lifetime of photogenerated intermediates [15,16]. However, we envisioned that employing a chiral auxiliary bound to the substrate could provide a straightforward and effective strategy for achieving stereocontrol in anion radical [2+2] photocycloaddition. In this study, we wish to present a simple approach for the synthesis of enantioenriched Bicyclo[3.2.0]heptanes exploiting the use of a chiral auxiliary.

## 2. Results

Previous studies have shown that chiral auxiliaries are an effective tool for controlling the stereoselectivity of photo-induced radical addition reactions, achieving enantioselectivity ratios of up to 83:17 [6]. Based on this approach, this research explores the use of a chiral auxiliary in developing a stereoselective version of the [2+2] photocycloaddition of aryl bis-enones. Due to their broad compatibility, easy preparation, and well-established effectiveness in controlling stereoselectivity, Evans oxazolidinones have been selected as chiral auxiliaries for the development of a stereoselective version of this reaction [17]. The process was carried out following a previously established general protocol, which involves the use of LiBr as a Lewis acid, Eosin Y as a photocatalyst, and visible light irradiation to promote the transformation [13]. The choice of LiBr was supported by previous studies demonstrating that coordinating cations can effectively influence the selectivity of [2+2] cycloadditions involving bis-enones, both by promoting substrate preorganization and by stabilizing key reactive intermediates [12,18].

Initially, bis-enones **7** and **8**, bearing different chiral oxazolidinones, were synthesized following the procedure outlined in Figure 2. (5*E*)-7-Oxo-7-phenylhept-5-enal (**3**) was obtained in 75% yield through the reaction of ylide **2** with glutaric aldehyde, whereas ylide **2** was prepared from commercially available 2-bromo acetophenone **1** using a standard Wittig reaction protocol [19].

Oxazolidinones **4a** and **4b** were then reacted with 2-chloroacetyl chloride, yielding compounds **5a** and **5b** in 65% and 85% yields, respectively. These intermediates were subsequently converted into the corresponding ylides **6a** and **6b** by treatment with PPh_2_ in CH_2_Cl_2_ for 48 h, followed by quenching with NaOH aqueous solution. Compounds **6a** and **6b** were then reacted with the previously synthesized compound **3** to afford enantiopure bis-enones **7** and **8** by treatment with MgSO_4_ in CH_2_Cl_2_ for 16 h. Compounds **7** and **8** were isolated in 70% and 50% yields, respectively, with the (*2E,7E*) diastereoisomer as the major product, which was easily separated through chromatographic purification.

To gain deeper insight into the redox properties of these substrates, cyclic voltammetry was performed on compound **7**. Two reduction peaks were observed at −0.85 V and −1.13 V vs. SCE. Notably, the reduction potential is more positive than that of photoexcited Eosin Y* (Eosin Y˙^+^ → ^3^EY* = −1.15 V vs. SCE) [20], indicating that compound 7 could be readily reduced under photoredox conditions. Moreover, electrochemical analysis of a symmetrical analogue revealed a cyclization process occurring within a potential range of −1.5 V to −2.0 V [8], suggesting that the [2+2] cycloaddition could proceed under these conditions (see Appendix A).

On the basis of these results, precursors **7** and **8** were subjected to the anion radical [2+2] photocycloaddition, mediated by LiBr as a Lewis acid, Eosin Y as a photocatalyst, and diisopropylethylamine (2 equiv) as both a base and quenching additive for photocatalyst regeneration. The reaction was carried out using a homemade photoreactor equipped with green LEDs as a light source operating at 530 nm, with an intensity of 424 mW/cm^2^ (see Appendix A). An initial screening was performed using both compounds **7** and **8**, revealing that the chiral auxiliary is compatible with the anion radical [2+2] photocycloaddition. Notably, although 16 different stereoisomers could theoretically be predicted (since the [2+2] cycloaddition generates a Bicyclo[3.2.0]heptane with four new stereocenters), only two of them were actually formed and isolated. The results are summarized in Table 1.

A solvent screening performed using compound **7** in the presence of 1 mol% Eosin Y as a photocatalyst identified dry acetonitrile as the optimal reaction medium. No product formation was observed when methanol or dichloromethane was used (entries 1 and 2), whereas the use of DMF or THF led to the formation of Bicyclo[3.2.0]heptanes **9a** and **9b** in 35% yield and 45% yield, respectively (entries 3 and 4).

Gratefully, the use of acetonitrile allows the desired product to be obtained in a 76% yield with a 60:40 diastereomeric ratio (entry 5). Reducing the catalyst loading to 0.5 mol% led to a slightly lower yield of 62% with a comparable diastereomeric ratio of 62:38 (entry 6). The replacement of Eosin Y with Na_2_Eosin Y instead proved to be detrimental for the reaction, since compound **9** was isolated in 23% yield only, along with a modest reduction in diastereoselectivity (entry 7).

Notably, compounds **9a** and **9b** were easily separated by chromatographic purification, and analysis of their coupling constants in ^1^H-NMR spectra allowed the determination that the anion radical [2+2] photocycloaddition selectively promoted the formation of the **9**-*cis-anti* cycloadducts [21]. This result differs from that observed in the anion radical [2+2] photocycloaddition of aryl bis-enones, where the *trans-anti* isomer was also detected [15].

A similar behavior was observed for compound **8**. When the reaction was performed using 0.5 mol% of photocatalyst, compound **10** was formed in 43% yield (entry 8), which increased to 57% when 1 mol% of Eosin Y was used (entry 9). It must be noted that in that case, the diastereoselection was higher compared to those obtained with compound **7**. As expected, decreasing the amount of LiBr proved detrimental to the reaction (entry 10) since it was already proven that the presence of a Lewis acid is essential for coordinating the two carbonyl groups of the aryl bis-enone during the anion radical [2+2] photocycloaddition [15]. To confirm the absolute configuration of the Bicyclo[3.2.0]heptanes formed, a pure sample of compound **9a** was crystallized from a mixture of hexane/diethyl ether (9:1), resulting in the formation of white crystals suitable for single-crystal X-ray diffraction studies. The absolute configuration of the product was unambiguously determined, by XRD analysis, to be (*1S,5R,6R,7S*)-9a, as shown in Figure 1. The absolute configuration of compound **9b** was then assigned by analogy on the basis of ^1^H-NMR coupling constants.

## 3. Reaction Mechanism

Computational analysis was then performed to further explore the regio- and stereoselectivity of this transformation. It has been well established that the anion radical [2+2] photocycloaddition of symmetric aryl bis-enones occurs through the photoexcitation of the photocatalyst upon light irradiation [10]. In its excited state, the photocatalyst is reduced by diisopropylamine, resulting in the formation of its radical anion. This reduced catalyst then interacts with the aryl bis-enone (**A**), activated by coordination with LiBr, generating the radical anion intermediate (**B**). At this stage, an intramolecular radical attack takes place, leading to the formation of the five-membered ring distonic radical **C**. This species then evolves into the bicyclic radical intermediate **D** through a subsequent intramolecular radical attack. Once compound **D** is formed, an oxidation reaction, promoted by the radical form of diisopropylamine, occurs, resulting in the formation of desired Bicyclo[3.2.0]heptane **E** (Figure 3).

This reaction mechanism could also be proposed for asymmetric bis-enones such as compounds **7** and **8**. However, with these derivatives, the loss of symmetry must be considered, as it can lead to the formation of two distinct distonic radicals, depending on which carbonyl group is involved in the first radical process. To investigate these hypotheses, we carried out a DFT study on the anion radical [2+2] photocycloaddition of the unsymmetrical bis-enone **8**. Initial geometries of compound **8** coordinated with two LiBr molecules were first generated through Monte Carlo conformational sampling, performed using the MMFFs force field [22] in the Macromodel package of the Schrödinger suite [23]. These structures were then fully optimized via DFT calculations, employing the unrestricted uM06-2X function [24] with the 6-31G(d,p) basis set in the Gaussian package [25]. To account for solvation effects, the PCM model [26,27] was used with acetonitrile as a solvent. The uM06-2X functional was chosen as it is one of the best-performing global hybrids, offering an improved description of long-range dispersive interactions compared to the B3LYP functional [28]. Local minima (characterized by the absence of imaginary frequencies) and transition states (identified by a single imaginary frequency) were determined through harmonic vibrational calculations.

After the interaction between compound **8,** LiBr, and the photocatalyst, two different radical anion intermediates, F and F’, could be generated, with the unpaired electron located either on the carbon in β-position relative to the carbamate moiety or on the carbon in β-position relative to the ketone moiety (Figure 4a). Unfortunately, spin density analysis was unsuccessful in determining which radical is preferentially formed, as the unpaired electron was found to be almost equally distributed between both regions. However, when the spin density distribution was calculated for the five-membered ring distonic radical **G**, formed via an intramolecular radical attack of compound **F** (Figure 4b), an increased density was observed on the β-carbon of the ketone moiety. This finding suggests that, even though both **F** and **F’** could be formed, only radical **F** is able to undergo the reduction step on the amide side, generating radical intermediate **G**, which subsequently evolves into the final bicyclic product through the mechanism depicted in Figure 4c.

Since compound **F** is generated in the presence of two molecules of LiBr, multiple coordination modes were computationally evaluated, as LiBr can interact with any of the three carbonyl groups present in compound **8**. Since this coordination flexibility can influence both the reaction pathway and the selectivity of the process, conformational analysis was performed revealing two preferred conformations: one in which two LiBr molecules act as a bridge, coordinating both C=O bonds simultaneously, and another in which each LiBr molecule coordinates a single carbonyl group. In the presence of the activated form of Eosin Y, these two structures underwent reduction, leading to the formation of the radical anions **F*_cis_***, where LiBr bridges both C=O bonds in a *syn* rearrangement, and **F*_trans_***, where bis-coordination occurs, resulting in an *anti* rearrangement of the C=O bonds. **F*_cis_*** and **F*_trans_*** were used as starting points for the DFT investigation of the reaction mechanism, and it was found that **F*_trans_*** is +8.9 kcal/mol disfavored compared to more stable **F*_cis_*** (Figure 5). Due to the presence of a chiral oxazolidinone group bound to the molecular structure, when the first intramolecular cyclization occurs in a *syn* or *anti* fashion, four different substituted cyclopentanes, **Ga–d**, can be generated. Intermediates **Ga** and **Gb** are derived from a *syn* attack, whereas **Gc** and **Gd** originate from an *anti* attack.

A notable difference in terms of energy has been observed between the *anti* closure pathway and the corresponding *syn* closure pathway (red line vs. black line in Figure 4).

As expected, **Gc** and **Gd** are higher in energy compared to **Ga** and **Gb**. The energy gap is even more evident when the related transition states are considered: **TS1**_(**F*cis***_**_→Ga)_** and **TS1**_(**F*cis***_**_→Gb)_**, responsible for the formation of **Ga** and **Gb**, respectively, are up to +3.7 kcal/mol higher in energy compared to initial compound **F*_cis_***. In contrast, **TS1**_(**F*trans***_**_→Gc)_** and **TS1**_(**F*trans***_**_→Gd)_**, responsible for the formation of **Gc** and **Gd,** respectively, show an energy gap of up to 27.9 kcal/mol compared to **F*_trans_***.

This analysis clearly demonstrates that the *anti* closure pathway is energetically less favorable compared to the more preferred *syn* closure pathway. As a result, the only products formed after the first radical closure are the intermediates **Ga** and **Gb**. This conclusion aligns perfectly with the experimental observations, since no product with the two hydrogens at *C_3_* and *C_7_* in the *anti* configuration was detected during this transformation. For these reasons, the *anti* closure pathway was not subjected to further computational investigation.

According to the proposed mechanism, when compound **G** is formed under aprotic conditions, a further intramolecular radical cycloaddition occurs, leading to the formation of bicyclic radical anion **H** by a formal [2+2] cycloaddition process. This newly formed radical anion is then oxidized to the corresponding Bicyclo[3.2.0]heptane product **I** by the presence of iPr_2_NEt ^•+^ (Figure 4c). Having fixed the stereocenter on the oxazolidinone moiety, eight different diastereoisomers of intermediate H are formed: **Haa, Hab, Hac**, and **Had** are derived from an intramolecular cyclization of structure **Ga**, whereas **Hba**, **Hbb**, **Hbc**, and **Hbd** are derived from a cyclization of structure **Gb**. According to the commonly accepted nomenclature for such products, where structures are identified based on the relative position of hydrogen atoms at *C_2_* and *C_7_* [15], **Haa** and **Hba** arise from a *cis-anti* [2+2] cycloaddition; **Hab** and **Hbb** result from a *cis-syn* closure; **Hac** and **Hbc** come from a *trans-syn* closure, and **Hab** and **Hbd** originate from a *trans-anti* closure.

All the energies associated with the radical intermediates **H_(a-d)(a-d)_** and the corresponding neutral products **I_(a–d)(a–d)_** were located, and the Gibbs free energy profile is reported in Figure 6. DFT calculation revealed that, among all the possible closure pathways, the most favorable one corresponds to the *cis-anti* closure. In particular, the transition states **TS2_(Ga__→Haa)_** and **TS2_(Ga__→Hba)_** were found to be +4.0 kcal/mol and +6.4 kcal/mol higher in energy compared to the **Ga** intermediate, respectively. Notably, these two transition states are the lowest in energy among all the investigated pathways, further supporting the preferential formation of the experimentally observed *cis-anti* products.

In addition, product **Iaa** is more stable than **Iba**, primarily due to the spatial arrangement of the phenyl group on the chiral oxazolidinone, which plays a crucial role in stabilizing the structure. In **Haa** and **Iaa**, the phenyl group remains positioned away from the aryl ketone moiety, minimizing steric interactions. Conversely, in **Hba** and **Iba**, the rigidity of the structure forces the phenyl group into closer proximity to the aryl ketone moiety, leading to increased steric hindrance (see Appendix A of Appendix A).

The results obtained from DFT calculations are in agreement with the experimental findings, which show that the only diastereoisomers observed at the end of the reaction correspond to those arising from the *cis-anti* closure pathway. Moreover, among these, compound **Iaa** corresponds to the major stereoisomer obtained, **10a**, as confirmed by X-ray analysis.

## 4. Conclusions

In conclusion, an enantioselective approach to the synthesis of Bicyclo[3.2.0]heptanes via organophotoredox-catalyzed radical anion [2+2] photocycloaddition has been developed. The use of chiral oxazolidinones bound to the bis-enone structures enabled the isolation of enantioenriched Bicyclo[3.2.0]heptane derivatives, demonstrating the potential of combining chiral auxiliaries with organophotoredox transformations to control stereochemical outcomes. Experimental and computational analyses confirm that the [2+2] cycloaddition promoted by Eosin Y in the presence of green light irradiation follows a *syn*-closure pathway, favoring the formation of *cis-anti* products, as observed through NMR and X-ray analysis. Additionally, DFT calculations provide valuable mechanistic insights into assessing the absolute configuration of the chiral Bicyclo[3.2.0]heptanes obtained. The alignment between theoretical predictions and experimental results validates the proposed reaction pathway and suggests potential strategies for further enhancing stereoselectivity in related photocatalytic transformations.

## 5. Materials and Methods

General procedure for the synthesis of Bicyclo[3.2.0]heptanes via anion radical [2+2] photocycloaddition. A dry 10 mL vial was charged with Eosin Y (1.35 µmol), LiBr (0.54 mmol, 2 equiv), dry acetonitrile (2.2 mL), and the desired aryl-enone (0.27 mmol, 1 equiv). The resulting solution was sonicated for 10 min, after which freshly distilled iPr_2_NEt (0.54 mmol, 2 equiv) was added. The reaction mixture was then degassed by three freeze–pump–thaw cycles. The vial was placed at a 0 cm distance from the surface of the photoreactor and irradiated with green LEDs at room temperature under continuous stirring for 4 h. After completion, the crude reaction mixture was directly loaded onto a silica gel column for purification.

## Data Availability

Data are contained within the article and Appendix A.

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
