# Peer review of "Organophotoredox-Catalyzed Stereoselective Synthesis of Bicyclo[3.2.0]heptanes via [2+2] Photocycloaddition"

_molecules, 2025, doi:10.3390/molecules30102090_

Round 1

Reviewer 1 Report

Comments and Suggestions for Authors

This manuscript describes the stereoselective synthesis of bi-cyclo[3.2.0]heptanes through a green-light-mediated EY-catalyzed [2+2] cycloaddition by using aryl bis-enone derivatives as the substrates. Two diastereomers were isolated in moderate yields with the cis-anti diastereoisomers as the major products, whose absolute configurations have been confirmed by XRD analysis. The proposed syn-closure pathway leads to the formation of cis-anti diastereoisomer. DFT calculations have also confirmed that this syn-closure pathway is favored due to a much lower transition state energy. The manuscript is well and clear-written.

It is a very interesting and sound story. However, experimental procedures for the synthesis of substrates 7 and 8, as well as all experimental data (e.g., NMR, HRMS, XRD) are missing. Therefore, I recommend publication of the work in Molecules with major revisions.

  1. Please upload a supporting information document with all the required data and experimental procedures.
  2. The authors should screen the solvents and see if the reaction yield and stereoselectivity could be improved. These results could be determined by crude NMRs directly.
  3. Some classic [2+2] cycloaddition literature should be cited (J. Am. Chem. Soc. 2004, 126, 6, 1634–1635; J. Am. Chem. Soc. 2008, 130, 39, 12886–12887).

Reviewer 2 Report

Comments and Suggestions for Authors

This manuscript presents a well-executed investigation into the stereoselective synthesis of bicyclo[3.2.0]heptanes through an anion radical [2+2] photocycloaddition of aryl bis-enone derivatives. The study effectively integrates both experimental and computational methodologies to elucidate the reaction mechanism and stereoselectivity. The research demonstrates innovation and is consistent with contemporary trends in sustainable photochemical synthesis. Below are detailed comments and suggestions for improvement.

  1. The author should provide characterization data and spectra for the synthesized compounds. The reviewer did not find the supplementary materials in the submission system.
  2. The author proposes LiBr as a Lewis acid and should reference relevant literature to support this claim. Furthermore, has the author conducted a screening of other Lewis acids?
  3. The authors need to investigate the redox potential of aryl bis-enone or provide literature data on the phases in order to elucidate the reaction mechanism.
  4. On Page 1, the author discusses the photocycloaddition of unsaturated enones. It is recommended that a schematic representation be included to enhance readers' comprehension of this section.

Reviewer 3 Report

Comments and Suggestions for Authors

Rossi et al. described the enantioselective synthesis of bicyclo[3.2.0]heptanes via an anion radical [2+2] photocycloaddition, using a chiral oxazolidinone auxiliary group. They conducted a small study with only two examples, using a previously reported method for the diastereoselective synthesis of structurally similar compounds. Additionally, they explored only two oxazolidinone derivatives, which did not exhibit complete diastereoselectivity. They also supported the experimental results with DFT calculations. This manuscript will be accepted after addressing a few minor comments:

 - Did the authors explore other Lewis acids or oxazolidinones? The authors observed that the diastereoselectivity increased with the phenyl derivative.

In pag 6, line 208: The authors mention a blue line in Scheme 4, should it be the red one instead?

 - Was the i-Prâ‚‚NEt redistilled or dried? Did the authors purchase it and use it as received?

 - In the compound characterization, all the 13C NMR data must have one decimal.

 - I will ask the author to add the structure of the corresponding compound to all the spectra, to help the readers and improve understanding.

Round 2

Reviewer 1 Report

Comments and Suggestions for Authors

I am satisfied with the revisions the authors made in the manuscript. I recommend its publication in Molecules directly.

Reviewer 2 Report

Comments and Suggestions for Authors

I would like to support the publication of this manuscript. I really like the added Scheme 1.